

# The OH (3-1) nightglow volume emission rate retrieved from OSIRIS measurements: 2001 to 2015

Anqi Li[1], Chris Z. Roth[2], Adam Bourassa[2], Douglas A. Degenstein[2], Kristell Pérot[1], Ole Martin Christensen[1], and Donal P. Murtagh[1]

[1]Chalmers university of technology, Department of Space, Earth and Environment, Gothenburg, Sweden
[2]Institute of Space and Atmospheric Studies, University of Saskatchewan, Saskatoon, Canada

**Correspondence:** Anqi Li (anqi.li@chalmers.se)

**Abstract.** The OH airglow has been used to investigate the chemistry and dynamics of the mesosphere and the lower thermosphere (MLT) for a long time. The infrared imager (IRI) aboard the Odin satellite has been recording the nighttime 1.53 μm OH (3-1) emission for more than 15 years (2001-2015) and we have recently processed the complete data set. The newly derived data products contain the volume emission rate profiles and the Gaussian approximated layer height, thickness, peak intensity and zenith intensity, and their corresponding error estimates. In this study, we describe the retrieval steps of these data products. We also provide data screening recommendations. The monthly zonal averages depict the well known annual oscillation and semi-annual oscillation signatures, which demonstrate the fidelity of the data set (https://doi.org/10.5281/zenodo.4746506, Li et al. (2021)). The uniqueness of this Odin-IRI OH long-term data set makes it valuable for studying various topics, for instance, the sudden stratospheric warming events in the polar regions and solar cycle influences on the MLT.

## 1 Introduction

The OH airglow is an important feature that aids us in understanding the thermal, dynamical and chemical variations that occur in the mesosphere and lower thermosphere (MLT) region. The infrared imager (IRI) of Odin-OSIRIS (Optical Spectrograph and InfraRed Imaging System) has routinely been measuring the 1.53 μm OH (3-1) emission since the launch in 2001 (nighttime measurements are only available between 2001 and 2015), but this complete data set has been processed only recently. In this study, we describe the retrieval of the volume emission rate (VER) profiles of the OH (3-1) layer, and characterise the layer in terms of peak intensity, peak height, thickness and zenith intensity.

The newly processed OH data set is unique. We can access the vertically resolved information thanks to the limb sounding geometry, which the ground-based OH observations such as those operating within the Network for the Detection of Mesospheric Change (NDMC) network can not provide. Several other space-borne instruments have also measured the limb profiles of the OH layers, namely the Improved Stratospheric and Mesospheric Sounder (ISAMS), the Global Ozone Monitoring by Occultation of Stars (GOMOS), the Microwave Limb Sounder (MLS), the Sounding of the Atmosphere by Broad-band Emis-





sionRadiometry (SABER), the Wind Imaging Interferometer (WINDII), the SCanning Imaging Absorption spectroMeter for Atmospheric CHartographY (SCIAMACHY) and the optical spectrograph (OS) of OSIRIS. Thanks to Odin's near-polar-orbit and the imaging technique, IRI provides excellent coverage over the high latitudes and high sampling frequency in the along-track direction. Besides, the surprisingly long lifespan of the mission has made it possible for the IRI instrument to collect a long-term data set which is the major advantage over the aforementioned space-borne observations, except for SABER which is still measuring.

The new IRI OH data set has the potential to provide insights into various important scientific topics. To briefly name a few, the short-term (in hours and days), mid-term (in months and seasons) and long-term (in years and decades) variations of the MLT chemical composition have been of interest in the scientific community for a long time. Small-scale gravity waves are known to be the driving force of thermal and dynamical structure in the MLT (e.g. Fritts and Alexander, 2003). The characterisation and quantification of these waves are essential to the modelling community to correctly predict the general behaviour of the region (e.g. Limpasuvan et al., 2016; Ern et al., 2018). The high sampling frequency of IRI is capable of resolving small scale coherent structures (on the order of 100 km) along track within the OH layer. Short-term anomalies in OH may also be induced by external forces such as solar energetic particles (SEPs) (e.g. Damiani et al., 2010), and Degenstein et al. (2005) demonstrated that the IRI oxygen channel has well captured such an event on October 28 2003.

Another application of this data set would be the study of sudden stratospheric warming (SSW) events particularly those resulting in an elevated stratopause (ES). SSWs are, in general, recognised by the weakening or even reversal of the westerly polar vortex caused by planetary-scale waves. Despite what the name suggests, SSWs are associated with cooling and upwelling in the mesosphere (e.g. Orsolini et al., 2017). Specifically Tweedy et al. (2013) showed that during such an event this led to an enhanced secondary ozone layer on a short-term scale. After several days from the onset, the recovery phase of ES-SSW is characterised by strong downwelling and warming which caused the episodic intensification of the OH emission observed at a lower altitude and can last for more than a month (mid-term scale anomalies) (Damiani et al., 2010; Shepherd et al., 2010; Gao et al., 2011; Dyrland et al., 2010). Sheese et al. (2014) briefly showed the intense OH (8-3) emission after the ES-SSW episode in 2008/2009 winter in their figure 7, recorded by the OS of OSIRIS. These ES-SSWs have occurred multiple times during the Odin mission as shown by Pérot and Orsolini (2021), which makes the new IRI OH data set well suited for investigating the thermal, dynamical and chemical couplings during these winters.

The global distribution of the semi-annual oscillation (SAO), annual oscillation (AO) and quasi-biennial oscillation (QBO) projected by the OH nightglow have been studied by Gao et al. (2010) using SABER, by von Savigny (2015) and Teiser and von Savigny (2017) using SCIAMACHY, by Shepherd et al. (2006) using WINDII and by Zaragoza et al. (2001) using ISAMS. The temporal and spatial coverage of the IRI OH data set can also be expected to demonstrate similar signatures, as seen in the OS OH nightglow observations (Sheese et al., 2014).

Even longer term variations that occur in the OH airglow can also be investigated, such as those associated with mesospheric cooling and the possible solar cycle influence. Clemesha et al. (2005) pointed out that we do not yet have a consistent picture of the long-term behaviour in the MLT regions concerning solar activity and possible human activities, noting that various measurement techniques have provided conflicting results. There have been several attempts to characterise the solar influence

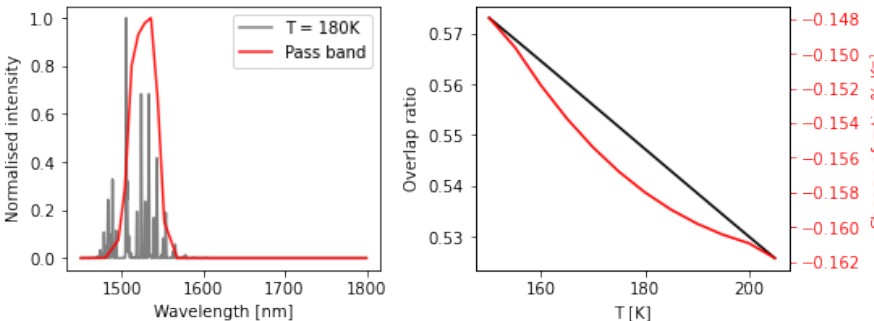

**Figure 1.** Left: the HITRAN modelled OH (3-1) spectrum at temperature of 180 K (grey) and IRI optical pass band (red). Right: The overlap ratio (black) of the OH (3-1) emission band (spectral interval 1.45-1.8 μm) as a function of temperature and its relative change (red).

using space-borne observations of OH emission (e.g. Gao et al., 2010; von Savigny, 2015; Teiser and von Savigny, 2017), but the data series available from those instruments were, at the time, hardly long enough to provide any concrete evidence. The
long life of the Odin spacecraft has made the IRI data record longer than the 11-year solar cycle, thus may be well suited for such investigations.

The structure of the paper is the following. Section 2 provides a brief description of the Odin-IRI instrument. Section 3 presents the method employed to derive the data products. The fidelity of the obtained data sets is demonstrated in Sect. 4, by looking at the SAO and AO signatures using the monthly zonal averages. Finally, conclusions are drawn in Sect. 5

## 65   2   Brief description of Odin IRI

Odin was launched into a polar, sun-synchronous orbit crossing the equator at 0600/1800 local time. The satellite orbits the Earth roughly 15 times per day. OSIRIS, one of the instruments aboard Odin, consists of two optically independent sub-instruments, the Optical Spectrograph (OS) and InfraRed Imager (IRI) (Llewellyn et al., 2004). The IRI instrument includes 3 channels operating at 1.530, 1.263 and 1.273 μm (channel 1, 2 and 3, respectively). Each channel contains an InGaAs array
detector with 128 pixels aligned vertically with 20 masked pixels at the lower end of the array for dark current correction. The recently updated calibration routines for the level 1 limb radiance data is essentially the same for all channels and they are described in Li et al. (2020), their section 2.1.

IRI channel 1 is designed to record both Rayleigh- and aerosol-scattered sunlight and the OH (3–1) Meinel vibration–rotation band airglow. As the recorded signal is dominated by the former in the day part and the latter in the night part of the orbit,
this study concerns only the measurements made during the night, i.e. solar zenith angle (SZA) larger than 90 degrees at the tangent point. The optical filter overlap on OH (3-1) emission band modelled using the HITRAN database (Gordon et al., 2017) (spectral interval 1.45 - 1.8 μm) is shown in Fig. 1 and overlap ratio is approximately 0.55 for emission temperatures between 150 and 200 K. The overlap ratio is insensitive to the rotational temperature of the emission (less than 0.2% change per K) and thus a single value is used through out this work.



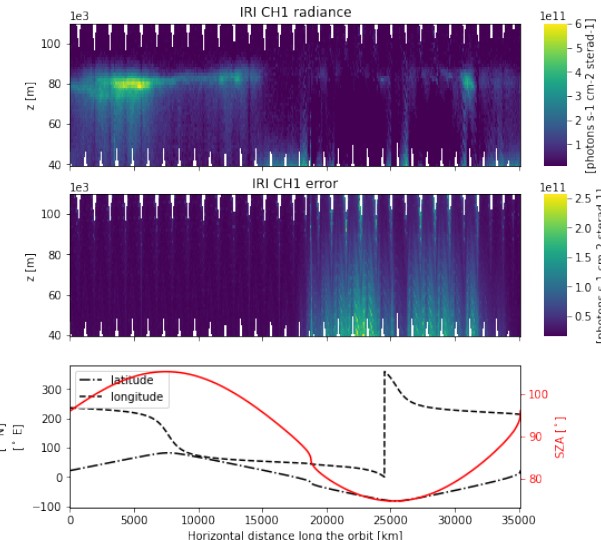

**Figure 2.** First and second rows: an orbit of IRI channel 1 limb radiance and its error estimate, respectively, recorded in orbit 10122. Last row: latitude (black dash-dot) and longitude (black dash) coordinate of the image represented tangent point as well as the solar zenith angle (SZA) at that location (red solid).

Figure 2 shows a typical orbit of limb radiance data (referring to the 'normal scanning mode' in Odin nomenclature). Due to the nodding motion of the Odin satellite, data plotted in altitude space shows data gaps in a 'zig-zag' pattern. SZA smaller than 90 degrees and/or pixels pointing below 60 or above 95 km tangent are not included in this study, as the signals are not dominated by the OH nightglow during these measurement conditions.

The number of orbits that contain at least one vertical limb radiance profile and at least one valid OH nightglow measurement are shown in Fig. 3. The inter-annual variability in data density is due to the differing operational conditions of Odin over the years. From 2001 to 2007, the observation schedule of Odin was shared equally in a 50% astronomy mode and a 50% aeronomy mode. For OSIRIS, this generally meant an arrangement of 3 days of continuous observation and 3 days of rest. From 2008, the astronomy part of the mission ended and OSIRIS made measurements all day every day for several years, thus the data density increased. In 2010, the first signs of ageing of the instrument and the satellite started to appear, meaning that 100% operation could not be maintained. Since then, various on/off patterns were arranged to maximise the number of measurements obtained while maintaining the health of OSIRIS and Odin as well as possible. Since 2016, OSIRIS can only be operated for a portion of every orbit (less than 50%, mainly making daytime measurements) due to a power supply unit problem, thus the OH nightglow measurements are highly limited, particularly after 2018.



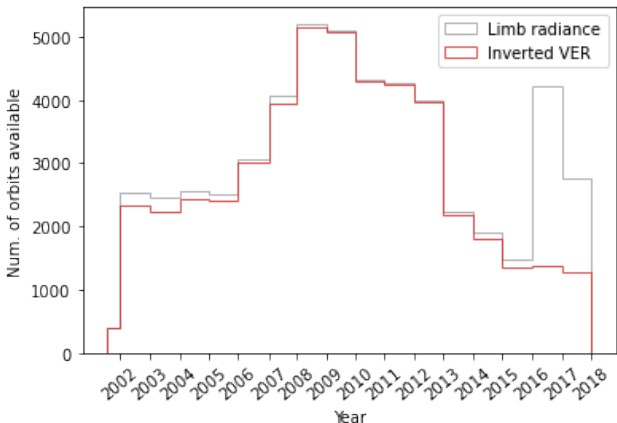

**Figure 3.** Data density from Odin launch until 2018. Number of orbits that contain at least one limb radiance profile (grey) and that contain at least one OH nightglow profile that is inverted to volume emission rate (VER) profile (red).

## 3 Data products

The main focus of this work includes the processing of two data sets, namely the VER of the OH (3-1) nightglow and the airglow layer characteristics in terms of zenith intensity, peak intensity, peak height and layer thickness. This section provides descriptions on how these parameters are derived in steps.

   The retrieval of the VER profiles follows the maximum a posteriori (MAP) estimation method (Rodgers, 2000). The principle equation of the inversion is

$$\hat{\mathbf{x}} = \mathbf{x_a} + \mathbf{G}(\mathbf{y} - \mathbf{K}\mathbf{x_a}), \tag{1}$$

$$\mathbf{G} = (\mathbf{K^T S_e^{-1} K} + \mathbf{S_a^{-1}})^{-1} \mathbf{K^T S_e^{-1}}, \tag{2}$$

where $\mathbf{x_a}$ and $\mathbf{S_a}$ are the mean and covariance, respectively, of the a-priori VER, $\mathbf{y}$ and $\mathbf{S_e}$ are the counterparts of the limb radiance measurement. $\hat{\mathbf{x}}$ is the inverted VER and $\mathbf{K}$ denotes the weighting functions that represents the linear relationship between the VER and limb radiance.

In this retrieval work, the atmosphere is approximated by a set of 1 km thick homogeneous layers ranging from 55 to 115 km, that is having a lower margin of 5 km below and a upper margin of 20 km above the limb radiance tangent range (60-95 km). The atmosphere is assumed optically thin for the emission, which means that the $\mathbf{K}$ matrix is essentially a representation of the path lengths for each line of sight through each atmospheric layer. $\mathbf{x_a}$ is set to be a zero vector and $\mathbf{S_a}$ is a diagonal matrix with constant value of $(1.1 \times 10^5)^2$ (photons s$^{-1}$cm$^{-3}$)$^2$ for diagonal elements representing altitudes between 60-95

km, and then exponentially decaying to zero for the grid points above and below this (illustrated by shadowing in Fig. 4 ). These a-priori values are used to be pragmatic as we avoid having to specify a particular vertical distribution for the layer thus complicating the later interpretation. We only regularise the airglow intensity in the form of the uncertainty of the zero a-priori



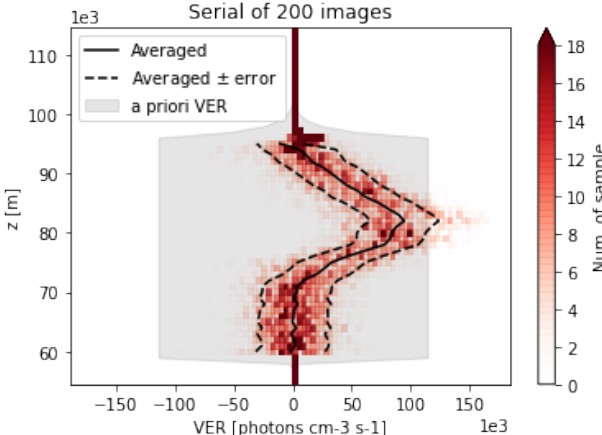

**Figure 4.** Histogram of the volume emission rate (VER) profiles retrieved from the first 200 images taken in orbit 10122 (see Fig. 7 for the whole orbit). Superimposed the averaged VER (solid line) and ± retrieval noise uncertainty (dashed lines) where the averaging kernel matrix maximum at row larger than 0.8 (see Fig. 5 . The a-priori mean (zero) and uncertainty is illustrated by the grey shadow.

profile to allow the inversion scheme to retrieve true to the observed emission with the boundary regions tapered to avoid edge effects.

The covariance of retrieval noise is

$$\mathbf{S_m} = \mathbf{G S_e G^T}. \tag{3}$$

Figure 4 depicts the retrieved VERs in form of 2D histogram of an arbitrary selected series of 200 images. The averaged VER and its error vector (i.e. square root of the diagonal elements in $\mathbf{S_m}$) of these sample profiles are good representations of a realistic OH airglow profile and its precision, respectively.

The averaging kernel (AVK) matrix

$$\mathbf{A} \equiv \frac{\partial \hat{\mathbf{x}}}{\partial \mathbf{x}} = \mathbf{G K} \tag{4}$$

represents the vertical resolution of the retrieved VER profile and maps the changes from the true state $\mathbf{x}$ to the estimated state $\hat{\mathbf{x}}$ at the corresponding atmospheric grid point. An example of the AVK matrix of a single retrieval is illustrated in Fig. 5. Where there are measurements available, the peak of the AVK is located near to the tangent grid point of the corresponding

measurement while for those AVKs representing higher altitudes the peak is close to 95 km. The full width half maximum of the peaks are 1 to 1.2 km between 60 and 95 km and rapidly increase to 2.4 km above 95 km, which represents the resolution of the retrieved VER profiles.

Once the VER profiles are obtained, the next step is to characterise the OH (3-1) emission layer in terms of the zenith intensity, peak intensity, peak height and layer thickness. A Gaussian model serves as a good approximation of the OH layer distri-

bution. As noted by Swenson and Gardner (1998), the modelled Gaussian parameters can be conveniently used for estimating

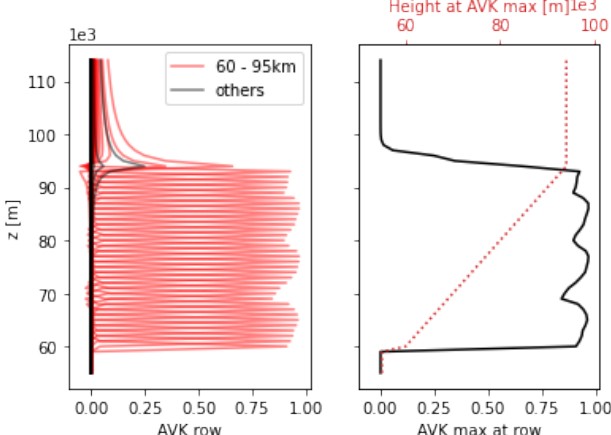

**Figure 5.** The averaging kernel (AVK) matrix of the first volume emission rate (VER) profile taken in orbit 10122 (see Fig. 7 for the whole orbit). Left: every row of the AVK matrix, with those representing the grid points between 60 and 95 km (red) and those below and above this range (black). Right: the maximum of the AVK matrix in each row (refers to the bottom axis) and the altitude where such maximum is found (refers to the top axis).

the relative intensity and rotational temperature fluctuations caused by a monochromatic gravity wave. Besides atmospheric wave-induced perturbations, not insignificant random noise is embedded in the signal due to the short integration time of IRI (about 1 s). We make use of the VER random error vector estimated in the previous step to perform a non-linear weighted least squares (WLS) fitting to a Gaussian model

$$V(z) = V_{peak} \cdot \exp(-\frac{(z - z_{peak})^2}{2\sigma^2}), \tag{5}$$

where the fitted result is the three Gaussian parameters: the airglow peak intensity $V_{peak}$, the peak height $z_{peak}$ and the full width half maximum (FWHM) layer thickness $2\sqrt{2\ln 2}\sigma$, along with their variances and co-variances. Only those VER values associated with a peak AVK value larger than 0.8 are taken as valid (see Sect.3.1). VER profiles having less than 10 valid grid points are then excluded from the Gaussian fitting procedure. An arbitrarily chosen VER profile is plotted along with the fitted Gaussian function in Fig. 6.

The fitted Gaussian parameters can then be used for further analysis. For instance, the integral of the fitted Gaussian function, which is denoted as the zenith intensity of the OH (3-1) layer $V_{zenith}$ in this study, can be computed as $\sqrt{2\pi} \cdot V_{peak} \cdot \sigma$, and its uncertainty follows the error propagation formula

$$e_{zenith}^2 = 2\pi(V_{peak}^2 \cdot e_\sigma^2 + \sigma^2 \cdot e_{peak}^2 + 2 \cdot V_{peak} \cdot \sigma \cdot e_{peak} e_\sigma \rho), \tag{6}$$

where the $e_{zenith}$, $e_{peak}$ and $e_\sigma$ denote the standard deviations of $V_{zenith}$, $V_{peak}$ and $\sigma$, and $e_{peak} e_\sigma \rho$ is the covariance between $V_{peak}$ and $\sigma$.

The night part of the corresponding orbit of the image in Fig. 6 is shown in Fig. 7. The reconstructed Gaussian layer represents well the overall morphology of the OH airglow, except that the asymmetry of the actual airglow layer, i.e. the

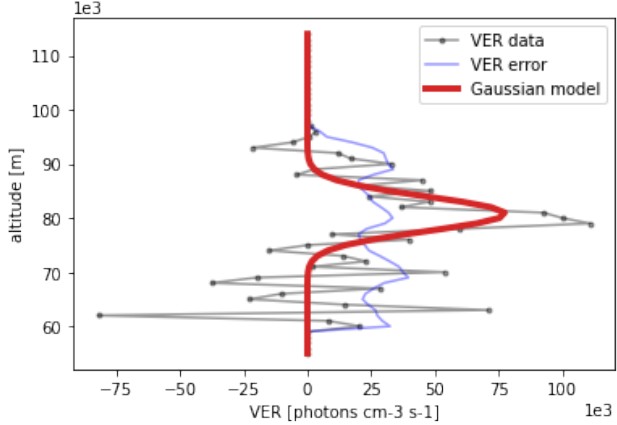

**Figure 6.** The fitted Gaussian model (red), the volume emission rate (VER) profile (grey) and the error profile of the VER (blue), which is the first profile taken in orbit 10122 (see Fig. 7 for the whole orbit). Note that only the grid points associated with an averaging kernel (AVK) maximum at row larger than 0.8 are used for the fitting, i.e. altitude of 60 to 95 km (refers to Fig. 5).

different vertical gradient at the lower and upper sides of the layer, cannot be reproduced by the Gaussian model. Also, when

the airglow appears to have a double-layered structure, the Gaussian model typically represents it as a very thick layer. Toward the right side of the plots, the airglow layer becomes very weak due to the twilight condition (SZA < 96 °), so that the Gaussian fitting finds large uncertainties for all parameters in that region. Where the airglow structure is prominent, the relative error in $V_{peak}, z_{peak}, 2\sqrt{2\ln 2}\sigma$ and $V_{zenith}$ are less than 25%, 2%, 35% and 60%, respectively, for this particular orbit.

The possible systematic error sources are summarised in Table 1. The absolute calibration of the radiance is estimated to

be the largest source of error since in-flight calibration capability is lacking. In order to investigate this uncertainty, we used a radiative transfer model (Bourassa et al., 2008; Zawada et al., 2015) to predict the twilight decay of the scattered sunlight signal at lower tangent altitudes, between 30 km and 40 km. These conditions are chosen because the upwelling radiation from the Earth surface and clouds is minimised during twilight, and 30-35 km is the upper reach of the stratospheric aerosol layer. Therefore, in these cases the limb signal is vastly dominated by Rayleigh single scattering and can be reliably predicted. Even

still there is a large amount of natural variability in the measured signal making a detailed absolute calibration very difficult to derive; however, we do not observe any systematic changes in the twilight decay of the measured signal when compared to the radiative transfer calculations over a large number of orbits spanning the mission lifetime.

The instrumental pointing is uncertain by 250 - 500 m (Llewellyn et al., 2004) and this translates to 10 - 15 % based on the OH gradient at the bottom and the upper sides of the layer. A small contribution of systematic error comes from the IRI

optical filter shape overlapping the OH (3-1) vibrational transition band assuming it is temperature insensitive. Moreover, the inversion scheme of VER assumes horizontally homogeneous layer cells. Any inhomogeneity will lead to an additional signal at a lower tangent altitude. In effect, the fitted Gaussian layer will become thicker. The associated systematic error depends on

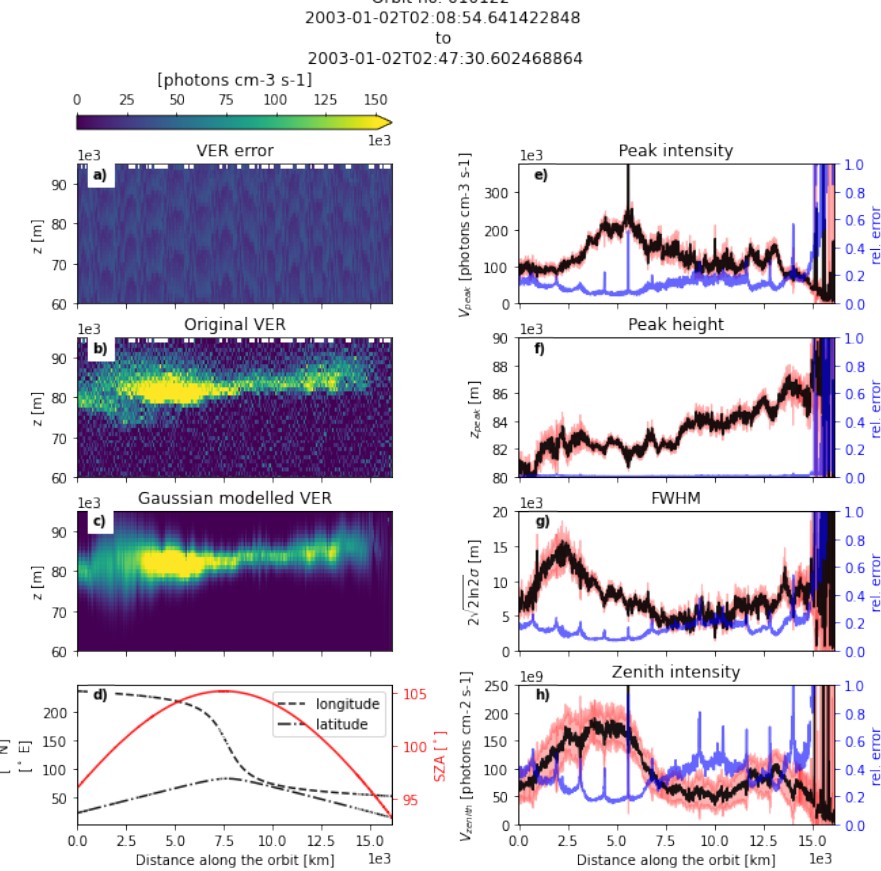

**Figure 7.** The resulted data products in orbit 10122. (a) The retrieved volume emission rate (VER) error, (b) the retrieved VER profiles, (c) the Gaussian reconstructed profiles, and (d) the longitude (black dashed) and latitude (black dash-dotted) coordinates of measurements along the orbit, and the solar zenith angle (SZA) (red solid) that refers to the right axis. The Gaussian modelled OH airglow layer (e) peak intensity $V_{peak}$, (f) peak height $z_{peak}$, (g) thickness $2\sqrt{2\ln 2}\sigma$ and (h) zenith intensity $V_{zenith}$, respectively. The black line corresponds to the expected value, the red shadow to the expected value plus minus the error and the blue line to the relative error which refers to the right axes.

how large and strong the structure is. Note that not all of these errors are relevant for all applications of this data set, and as such the user need to reflect on which systematic errors apply for their particular usage.

## 3.1 Data screening recommendations

To perform meaningful analyses, the data set must be screened as a first step. Because we used an arbitrarily chosen prior information about the VER profile, estimated VER grid points with low peak values or low sum of the rows in the AVK matrix, i.e. low measurement response (MR), should not be included in the analysis. A general rule of thumb is that values lower than





**Table 1.** A summary of the possible sources of systematic error in the IRI OH VER and their relative errors.

| Error sources | Estimated errors |
|---|---|
| Absolute calibration[a] | <20% |
| Instrumental pointing | 10-15%[b] |
| Temperature dependent filter overlapping | <1.6%[c] |
| Non-horizontally homogeneity effect | see text |

a: due to lack of in-flight calibration

b: estimated from 250 - 500 m bias in the pointing uncertainty

c: estimated from 10 K change around the mesopause

0.8 shall be excluded (Rodgers, 2000). This type of screening typically means that the VER profiles are shortened to the altitude
grid where measurements exist. The data gaps created are most significant when Odin executes the so-called 'mesospheric scan
mode' required by other instruments on board. In such cases the lowest IRI pixel can be as high as 90 km (see Fig. 4 in Li et al.
(2020) for reference). We do not perform the Gaussian fitting from a VER profile where the range does not cover at least the
75 to 88 km altitude range.

Another consideration is associated with twilight conditions, at the relevant altitude range this is when SZA is between
90-99.8°. As the OH (3-1) dayglow intensity is three times weaker than the nighttime counterpart (Llewellyn et al., 1978), the
emission measured under twilight conditions could differ from that during the night. However, if we screen out all measure-
ments that were taken with SZA<99.8 ° condition, the majority of the data recorded around the equatorial region all-year-round
as well as the mid-high latitudes around the equinoxes will be removed. This is a result of the Odin 6-18h orbit. However, the
diurnal variation of OH (3-1) is mainly associated with ozone photodissociation in the Hartley band, which is not significantly
affected until the sun rises above a SZA of 96 °. Therefore, in order to maximise the available data a lower limit of SZA to 96
° may be used as a screening policy.

Additionally we generally recommend that the user screens particularly noisy profiles. This can be done by either looking
at the retrieval error ('error2_retrieval') or cost function ('chisq') of the retrieved profile. The exact filter thresholds use will
depend on the application and thus the end user will need to determine this appropriate value. Similarly if the Gaussian
parameters are used, outliers should be filtered on the total cost of the Gaussian fit ('chisq').

## 4   Zonal averages

This section provides a demonstration of the consistency of the new IRI OH data set. The main purpose is to demonstrate the
seasonal variability, known from other studies of OH* such as Shepherd et al. (2006) using WINDII, Gao et al. (2010) using
SABER, Sheese et al. (2014) using OS and Teiser and von Savigny (2017) using SCIAMACHY. Quantitative and detailed
analyses of the characteristics and causes of these variations is beyond the scope of this paper.


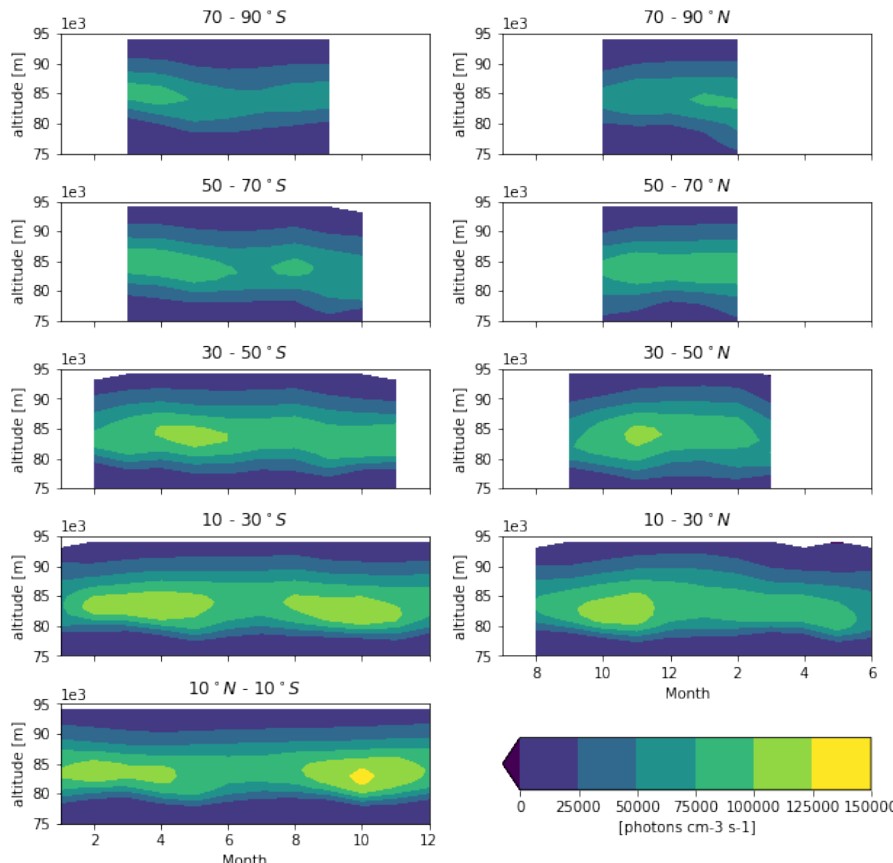

**Figure 8.** Monthly mean zonal average volume emission rate (VER) of all available years. Note: northern hemisphere plots have been shifted by 6 months to ease comparison between the hemispheres.

Figure 8 shows the monthly zonal mean (20° latitude bins) of the raw VER profiles for all years where data are available, following the screening policy recommended in Sect. 3.1. Monthly means for each year are computed first then averaged to a general monthly mean product to prevent the years with higher sampling density from dominating the average. Note that the months in the northern hemisphere (NH) are shifted by 6 months in Fig. 8 so that the colour contours are centred on the plots to ease the comparison between the hemispheres. The data covers nearly the whole year between 30 °N to 30 °S, and only the winter months at high latitudes.

At mid-low latitudes, the two maxima at the equinoxes are clearly visible and they weaken towards the poles, which represents the SAO signature. The two maxima are relatively more symmetric at 10 - 30 °S while this is not the case in the equatorial belt and 10 - 30 °N, which is also seen in SABER measurements (Gao et al., 2010) and SCIAMACHY measurements (von Savigny, 2015). In contrast to Shepherd et al. (2006), Gao et al. (2010) and von Savigny (2015), the autumn maximum is stronger than the spring maximum in NH mid-low latitudes. The airglow is generally brighter in the equatorial region than in

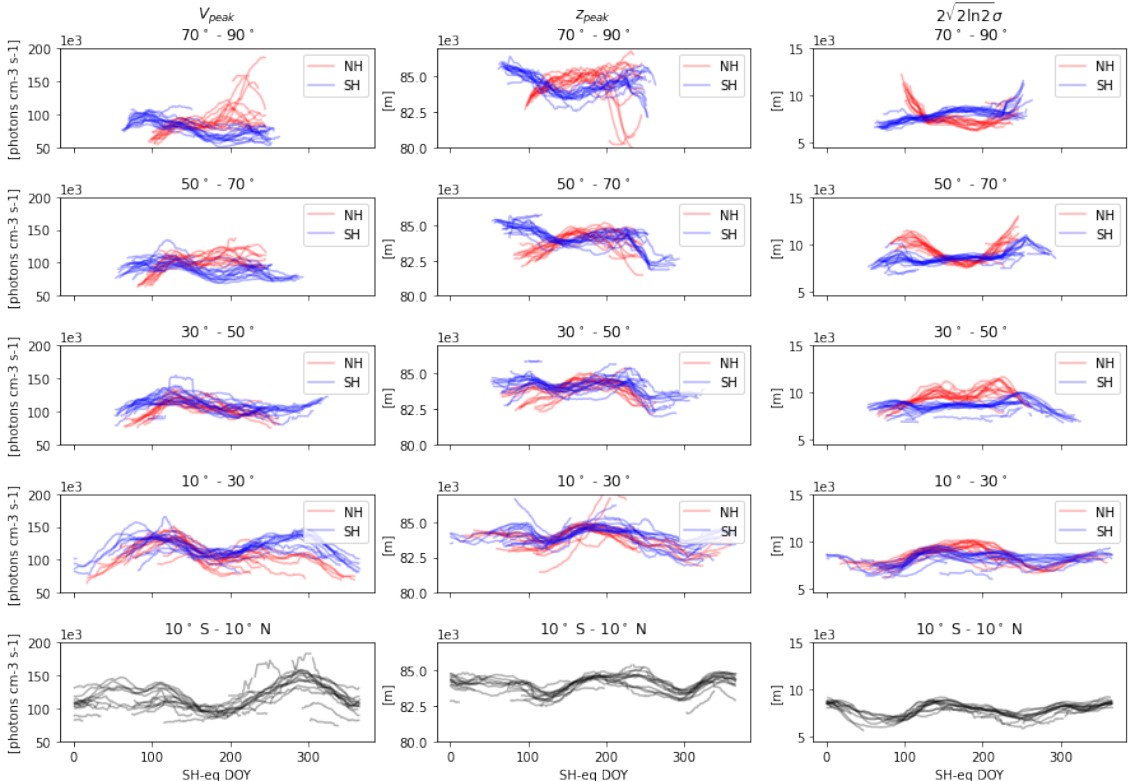

**Figure 9.** Zonal average monthly running means of the airglow peak intensity (first column), peak height (second column) and layer thickness (third column). Each line represents one year of data binned in 20 ° latitude bins labelled on top of each panel. Note that the northern hemisphere (NH) plots (in red) are shifted with 180 days to centre the winter periods to those of the southern hemisphere (SH) (in blue) and the tropics (in black, bottom row), thus the horizontal scale is labelled as SH equivalent (SH-eq.) day of the year (DOY).

the polar regions, which agrees with SABER (Baker et al., 2007) and SCIAMACHY measurements (Teiser and von Savigny, 2017).

At high latitudes above 50°, the SAO signature weakens. The lack of data in the summer months makes difficult to provide
clear evidence of the AO variation shown in, for example, Gao et al. (2010). The gradient of the bottom side of the airglow layer is less steep (i.e. the layer is more symmetrical around the peak) in comparison to the counterpart at lower latitudes. The hemispheric anti-symmetry between the two poles is also visible. In the southern hemisphere (SH), the winter airglow begins with a brighter and higher structure, then becomes less bright moving to a lower altitude in the middle of winter, and finally moves back to a higher altitude while keeping its weaker brightness at the end of the winter. In the NH winter, the
airglow peak altitude is rather flat throughout the winter, except in the early spring the bottom side becomes extremely thick which is associated with the several episodic SSW events that happened during the mission. The differences between the two winter-time polar regions are related to the varying strength of the polar vortices.





Figure 9 depicts the monthly running averages (30-day-running window) of the three Gaussian parameters, binned in the same latitude zones as those in Fig. 8 except that we show averages for every year available. Similarly to Fig. 8, the SAO
signature are evident in all three parameters especially at the lower latitudes. The SAO signature in peak height at low latitudes has maxima at solstices (minima at equinoxes), which is also shown by Sheese et al. (2014), Gao et al. (2010), Teiser and von Savigny (2017) and Wüst et al. (2020). Figure 9 also shows the anti-correlation between the peak intensity and peak height which is well studied by, among others, Liu and Sheperd (2006); von Savigny et al. (2012). The anti-correlation also exists between the peak intensity and layer thickness, which is relatively less studied by others. The hemispheric asymmetry is also
visible in the high latitude bands, as already shown in Fig. 8, since they are closely related to the different characteristics of the two winter polar vortices. Note that discrepancies in observational patterns by the satellite around the north and south poles also exist. The hemispheric asymmetry is also analysed and discussed by Gao et al. (2010), but their focuses were on mid-low latitudes rather than high latitudes (up to 50 °). In Sheese et al. (2014), we can also infer from their Fig. 5 the hemispheric asymmetry at higher latitudes. Several anomalies seen around the north pole are associated with the episodic SSW events that
result in a very intense airglow layer centred at a much lower altitude with a greater thickness, which is depicted in Winick et al. (2009), Damiani et al. (2010), Gao et al. (2011) and Sheese et al. (2014). In general, the variations depicted by these averages of the Gaussian fitted parameters are consistent with the layer characteristics shown in Fig. 8, thus the data set is self-consistent.

In summary, the IRI OH data set, including the VER profiles and the layer characteristics, is generally consistent with other
data sources while some differences are obviously present. These could be due to local time and other sampling issues and would require a detailed study to resolve.

## 5  Conclusions

The 15-year-data set of the OH (3-1) nightglow measurements collected by the infrared imager (IRI) of Odin-OSIRIS has been recently processed. The data set includes two bundles of products: the volume emission rate (VER) profiles and the
characteristics of the OH layer in terms of the peak intensity, peak height, layer thickness and zenith intensity. The VER profiles are retrieved by using the maximum a posteriori (MAP) method assuming horizontally homogeneous atmospheric layer cells. The layer characteristics are assessed by fitting a Gaussian model to the valid VER grid points using the nonlinear weighted least square (WLS) method. All data products contain their error estimations. The recommended data screening policy is described in this paper.
The zonal averages of the data products depict the seasonal variations of the OH layer, consistent with previous studies. These are the well known semi-annual oscillation (SAO) and annual oscillation (AO) signatures illustrating the fidelity of the data set. The new IRI OH data set is hereby made available to the scientific community, and has good potential for studying several topics in the future such as the effect of sudden stratospheric warming (SSW) events and solar cycle influences in the mesosphere and the lower thermosphere (MLT) region, thanks to the coverage over the high latitudes and the long lifespan of
the Odin satellite.



## 6 Data availability

The IRI OH (3-1) nightglow volume emission rate and its characteristics data sets are made publicly available and can be downloaded from the Zenodo data centre at https://doi.org/10.5281/zenodo.4746506 (Li et al., 2021).

Data sets are structured in one year per file in NetCDF format. File names are in the format of 'iri_ch1_ver_(year).nc', such as 'iri_ch1_ver_2008.nc' for data collected in 2008. The individual parameters included are described in Table 2 in detail.

*Author contributions.* The main author has prepared all the calculations and figures, the University of Saskatchewan authors (CR, DD, AB) have produced the calibrated IRI data. All authors (AL, CR, DD, AB, OLM, KP, DM) have contributed to the discussions.

*Competing interests.* The authors declare that they have no competing interests.

*Acknowledgements.* Odin is a Swedish-led satellite project funded jointly by the Swedish National Space Agency (SNSA), the Canadian
Space Agency (CSA), the National Technology Agency of Finland (Tekes), and the Centre National d'Etudes Spatiales (CNES) in France. Odin is also part of the ESA's third party mission programme.





**Table 2.** Parameters included in 'iri_ch1_ver_(year).nc' files, where '(year)' should be replaced with a four-digits-number representing the year.

| parameter name | units | type | dimension | description |
|---|---|---|---|---|
| time | - | datetime64 | (time) | Time at the beginning of the IRI exposure |
| z | m | float32 | (z) | Altitude grid of VER retrieval |
| latitude | degrees N | float32 | (time) | Latitude at the tangent point |
| longitude | degrees E | float32 | (time) | Longitude at the tangent point |
| orbit | - | int32 | (time) | Orbit number of Odin when data was collected |
| sza | degrees | float32 | (time) | Solar Zenith Angle between the satellite line-of-sight and the sun |
| apparent_solar_time | hour | float32 | (time) | Apparent Solar Time at the line-of-sight tangent point |
| ver | photons $\mathrm{cm}^{-3}\mathrm{s}^{-1}$ | float32 | (time, z) | IRI OH(3-1) volume emission rate |
| mr | - | float32 | (time, z) | Measurement response |
| A_diag | - | float32 | (time, z) | Averaging kernel matrix diagonal elements |
| A_peak | - | float32 | (time, z) | Averaging kernel maximum in each row |
| A_peak_height | m | float32 | (time, z) | Corresponding altitude of the averaging kernel maximum in each row |
| error2_retrieval | $(\mathrm{photons\ cm}^{-3}\mathrm{s}^{-1})^2$ | float32 | (time, z) | Retrieval noise $S_m$ diagonal elements (Rodgers (2000)) |
| error2_smoothing | $(\mathrm{photons\ cm}^{-3}\mathrm{s}^{-1})^2$ | float32 | (time, z) | Smoothing error $S_s$ diagonal elements (Rodgers (2000)) |
| peak_intensity | photons $\mathrm{cm}^{-3}\mathrm{s}^{-1}$ | float32 | (time) | Peak intensity (Gaussian fit) |
| peak_intensity_error | photons $\mathrm{cm}^{-3}\mathrm{s}^{-1}$ | float32 | (time) | Error of peak intensity (Gaussian fit) |
| peak_height | m | float32 | (time) | Peak height (Gaussian fit) |
| peak_height_error | m | float32 | (time) | Error of peak height (Gaussian fit) |
| peak_sigma | m | float32 | (time) | Layer thickness $\sigma$ (Gaussian fit) |
| peak_sigma_error | m | float32 | (time) | Error of layer thickness $\sigma$ (Gaussian fit) |
| zenith_intensity | photons $\mathrm{cm}^{-2}\mathrm{s}^{-1}$ | float32 | (time) | Integration of the Gaussian function |
| zenith_intensity_error | photons $\mathrm{cm}^{-2}\mathrm{s}^{-1}$ | float32 | (time) | Error of the Gaussian integration |
| cov_peak_intensity_peak_height | photons $\mathrm{cm}^{-3}\mathrm{s}^{-1}\mathrm{m}$ | float32 | (time) | Error covariance of peak_intensity and peak_height |
| cov_peak_intensity_peak_sigma | photons $\mathrm{cm}^{-3}\mathrm{s}^{-1}\mathrm{m}$ | float32 | (time) | Error covariance of peak_intensity and peak_sigma |
| cov_peak_height_peak_sigma | $\mathrm{m}^2$ | float32 | (time) | Error covariance of peak_height and peak_sigma |
| chisq | - | float32 | (time) | $\chi^2$ cost of the Gaussian function fitting |



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
