# Peer review of "The OH (3-1) nightglow volume emission rate retrieved from OSIRIS measurements: 2001 to 2015"

_Earth System Science Data, 2021_

## Author Response (AR1)

**AC-ESSD**

RC #1

This work presents a valuable data set of the nighttime 1.53 μm OH(3-1) emission observations measured for more than 15 years (2001-2015) by the infrared imager (IRI) aboard the Odin satellite. Information about volume emission rate (VER) profile of the OH emission layer, peak height and peak intensity are retrieved from the limb radiances measured. I am sure that this set of data will help to improve the knowledge of OH emission global behaviour. On the other hand the paper is clear, well written, and the figures and tables help to clarify the paper.

**We would like to thank referee #1 for taking the effort to review this manuscript. Our replies to your comments as well as some modification to the manuscript are provided below.**

General remarks:
Satellite measurements have a global spatial coverage and produce very important global information. In this sense this set of IRI measurements together the obtained in a similar temporal interval by SABER, will help to study globally OH emission behaviour.

In the paper there is a brief comment (lines 19-20) about ground-based observations that reflects that they, certainly, do not have vertical resolution. However I would like the relevance of ground-based observations to be pointed out in the paper. Due to their high sampling frequency and temporal coverage at their locations, there are important long data sets of ground-based OH observations that are producing important information about the behaviour of OH emissions. This deserves be mentioned. Reisin and Scheer (2017) have showed an interesting comparison between ground-based and satellite airglow measurements. The authors address a difference in SABER temperatures while the satellite overpasses at El Leoncito (from the East or from the West) by comparing SABER data to those they found in their Argentine Airglow Spectrometer (AAS) ground-based data, explaining these differences as result of both: the large amplitude of the semidiurnal tide and the differences in the temporal sampling of SABER. Hopefully, satellite and ground-based data sets used together will help to improve our knowledge about OH emission behaviour.

**We have added the sentence**
**"However, ground-based measurements have advantages over satellite observations, because they provide data sets with better temporal coverage, both in terms of length and frequency, at a given location."**
**to acknowledge the ground-based observations do have advantages that space-borne observations can not offer. Indeed, the combination of different data sets will help us to understand the OH emission better.**

In the description of the Data products, the Figure 6 is the one that best explains the retrieved VER profiles from IRI data and the modelled OH airglow layer characteristics. This figure shows a typical VER profile retrieved from orbit 10122. Large uncertainties can be seen from 60-79 km that can reach values of 25-50 1e3 photons cm-3s-1 (50% of the peak emission value) and above 89 km can reach values of about 25 1e3 photons cm-3 s-1 (25% of the peak emission value).

We have added the following sentence to point out the quantification of the VER error in percentage value, after Figure 4 where 200 profiles, instead of one profile in Figure 6, are represented.
"The averaged uncertainty of VER is about 30% relative to the airglow peak."

On the other hand the IRI retrieved profile reaches the emission peak at height at least 2 points lower (in the figure) than the modelled Gaussian profile (that means 2-3 km lower). These errors in the retrieved and modelled VER profile should be quantified and discussed.

In terms of the Gaussian approximated parameters, the uncertainties are quantified in the dataset. For the profile shown in Figure 6, the uncertainty of the layer peak height is 563 m. The whole orbit is visualised in Figure 7. The red shadow in panel (f) indeed shows a roughly 2-3 km uncertainty at the beginning of the orbit due to the low intensity or a double-layered structure (non-Gaussian structure).
The following sentence is added under the caption of Figure 6.
"The fitted Gaussian parameters are V_peak= 7.76 (±1.51)×10^4 photons cm−3 s−1, z_peak= 80.8×10^3 (±563) m, σ= 3.2×10^3 (±581) m."
And a sentence is added to the text after Figure 7.
" The non-Gaussian structure of the airglow layer is represented in the error estimation of the peak height and thickness."

In data screening recommendations, I am afraid I am far from the field and I do not understand the general rule of values lower than 0.8 (of what? from peak values? from measurement response?). Could this be rewritten in more detail?

The definition of averaging kernel (AVK) matrix is given after Eq. (4). It maps the changes from the true state to the estimated state at corresponding altitudes. Since each row of AVK is close to a perfect delta function, the sum of each row (measurement response) is essentially the same as the peak of each row. These values indicate how sensitive is the retrieved VER to the true state -- if the true state changes by 1 unit and the corresponding retrieved VER changes by 0.8 unit, it is generally considered as a "good" retrieval. In practice, we used both MR and peak AVK conditions to screen out data in the following sections.
The following sentence has been added to the data screening recommendation section, in order to clarify that point:
"A rule of thumb is that values of MR and/or peak of AVK lower than 0.8 shall be excluded (Rodgers, 2000) since the retrieved VER is, in such cases, insensitive to the true state (i.e. equal to the zero a priori). "

Section 4 provides a demonstration of the consistency of the IRI OH data. IRI OH seasonal variability is compared with the obtained in other studies (Shepherd et al., 2006; Gao et al-, 2010; Sheese et al-, 2014; Teiser and von Savigny (2017), however I notice that the work of Garcia-Comas et al., 2017 is missed. This work uses OH SABER data to analyse the variability of OH emission, OH rotational temperature and altitude of the OH layer at mid-latitudes. It would be interesting to compare the results obtained from IRI with those obtained from SABER at mid-latitudes.

We acknowledge your recommendation and added the reference to Section 4, as well as to the introduction section as the following:
"Garcia-Comas et al., 2017 used the SABER measurements to derive an empirical formula in order to predict the emission altitude of the OH layer measured over the Sierra Nevada Observatory in Granada. They found a

**significant short-term variability caused by overlapping waves in the scale of few hours."**

RC #2

The manuscript describes the retrieval and validation of a new data product, OH(3-1) nightglow volume emission rates, from the Infra-Red Imager on board the Odin-OSIRIS satellite. The dataset covers a period of 15 years and can facilitate the investigation of the dynamic variability of the upper mesosphere/lower thermosphere (MLT) region over a full solar cycle. In combination with other correlative in time ground-based and satellite observations, these long time series data can yield a very valuable and unique information for the understanding the effect of various neutral dynamics and solar influences on the MLT region.

I recommend publication in its present form.

**We would like to thank referee #2 for taking the effort to review this manuscript and gave a positive comment.**